Different patterns of foreground and background processing contribute to texture segregation in humans: an electrophysiological study

Zhang Baoqiang 1 2 3
Hu Saisai 1 2 3
Zhang Tingkang 1 2 3
Hai Min 1 2 3
Wang Yongchun 1 2 3
Li Ya 1 2 3 yali@snnu.edu.cn
Wang Yonghui 1 2 3 wyonghui@snnu.edu.cn
1 School of Psychology, Shaanxi Normal University , Xi’an , China
2 Shaanxi Provincial Key Laboratory of Behavior & Cognitive Neuroscience , Xi’an , China
3 Shaanxi Provincial Key Research Center of Child Mental and Behavioral Health , Xi’an , China
Fogt Nick
Electronic publication date: 2023 Oct 2
Publication date: 2023
Volume: 11
Electronic Location ID: e16139
Received 2022 Oct 11; Accepted 2023 Aug 29
Copyright: © 2023 Zhang et al.
Copyright year: 2023
Copyright holder: Zhang et al.
License: This is an open access article distributed under the terms of the Creative Commons Attribution License, which permits unrestricted use, distribution, reproduction and adaptation in any medium and for any purpose provided that it is properly attributed. For attribution, the original author(s), title, publication source (PeerJ) and either DOI or URL of the article must be cited.
License URL: https://creativecommons.org/licenses/by/4.0/

Keywords: Figure-ground organization, Texture segregation, Temporal response function (TRF), Electroencephalogram (EEG)

Funding: National Natural Science Foundation of China 31800910 This work was supported by the National Natural Science Foundation of China (No. 31800910). The funders had no role in study design, data collection and analysis, decision to publish, or preparation of the manuscript.

==============================
Background

Figure-ground segregation is a necessary process for accurate visual recognition. Previous neurophysiological and human brain imaging studies have suggested that foreground-background segregation relies on both enhanced foreground representation and suppressed background representation. However, in humans, it is not known when and how foreground and background processing play a role in texture segregation.

Methods

To answer this question, it is crucial to extract and dissociate the neural signals elicited by the foreground and background of a figure texture with high temporal resolution. Here, we combined an electroencephalogram (EEG) recording and a temporal response function (TRF) approach to specifically track the neural responses to the foreground and background of a figure texture from the overall EEG recordings in the luminance-tracking TRF. A uniform texture was included as a neutral condition. The texture segregation visual evoked potential (tsVEP) was calculated by subtracting the uniform TRF from the foreground and background TRFs, respectively, to index the specific segregation activity.

Results

We found that the foreground and background of a figure texture were processed differently during texture segregation. In the posterior region of the brain, we found a negative component for the foreground tsVEP in the early stage of foreground-background segregation, and two negative components for the background tsVEP in the early and late stages. In the anterior region, we found a positive component for the foreground tsVEP in the late stage, and two positive components for the background tsVEP in the early and late stages of texture processing.

Discussion

In this study we investigated the temporal profile of foreground and background processing during texture segregation in human participants at a high time resolution. The results demonstrated that the foreground and background jointly contribute to figure-ground segregation in both the early and late phases of texture processing. Our findings provide novel evidence for the neural correlates of foreground-background modulation during figure-ground segregation in humans.

Introduction

Figure-ground organization segregates an object (the foreground) from the background in a visual scene based on textural differences. It is a critical process for visual identification. Foreground-background segregation occurs when visual gradients of local features appear. These gradients may appear as an image’s orientation, luminance, or motion or through temporal cues (Bach & Meigen, 1992, 1997; Bach et al., 2000; Kandil & Fahle, 2001; Lamme, Van Dijk & Spekreijse, 1992). The ability to segment a whole figure into foreground and background parts has a profound influence on perception since the image elements of a figure are preferentially processed and leave stronger memory traces (Baylis & Cale, 2001; Driver & Baylis, 1996). It has long been theorized that foreground-background segregation has a crucial role as the initial step in visual recognition (Biederman, 1987). The neural correlates of texture segregation are of great interest for researchers.

Foreground-background segregation has been extensively studied with electroencephalogram (EEG) recording in humans and with multi-electrode recordings in monkeys. In humans, visual evoked potentials (VEPs) have been used to study foreground-background texture segregation with high temporal resolution. The VEPs of homogenous textures are subtracted from texture-defined figures to extract segregation-specific components, resulting in texture segregation VEPs (tsVEP). A classical EEG study in humans found that tsVEP revealed negative deflection in the occipital electrodes at about 160–225 ms (Bach & Meigen, 1992). The peak latencies for the main components of tsVEP differed from 110 to 300 ms in the posterior electrodes within and between studies (Bach & Meigen, 1997; Caputo & Casco, 1999; Fahle et al., 2003; Heinrich, Andres & Bach, 2007). These VEP studies suggested that the occipital region was involved in foreground-background segregation. Neurophysiological studies have been used to confirm these results in humans. In monkeys, Lamme (1995) found that neural activity at V1 was enhanced when the receptive field (RF) was placed in the foreground relative to the background. This effect is known as figure-ground modulation (FGM). The FGM had a delay of 30–40 ms relative to the onset of the first visually-induced activity. Many neurophysiological studies also found FGM in early visual areas (Lamme, 1995; Lamme, Rodriguez-Rodriguez & Spekreijse, 1999; Marcus & Essen, 2002; Rossi, Desimone & Ungerleider, 2001; Zipser, Lamme & Schiller, 1996). However, these earlier studies could not determine whether figure-ground modulation was caused by neural responses to the foreground, the background, or both.

A recent neurophysiological study investigated this question by adding a neutral condition. They compared the neural responses of the V1 and V4 sites to figure textures with uniform textures, and found that both the enhanced foreground activity and suppressed background activity contributed to FGM in monkeys (Poort et al., 2016). Human brain imaging studies have also suggested foreground modulation and background modulation during texture segregation. Using functional magnetic resonance imaging (fMRI), Papale et al. (2018) have provided evidence for enhanced foreground representation in both low- and high-level visual areas, and suppressed background representation at V4 and in the lateral occipital complex (LOC) during the passive viewing of natural images. Additional study has supported foreground enhancement and background suppression at V1 with behavioral and BOLD signal measurements (Huang et al., 2020). However, it is difficult to infer the temporal dynamics of foreground and background modulation during texture segregation using fMRI because of limited temporal resolution.

Previous foreground-background studies mainly focused on the occipital regions of the brain (Bach & Meigen, 1992, 1997; Caputo & Casco, 1999; Heinrich, Andres & Bach, 2007), while more recent research has attempted to determine whether high-level areas or regions are involved in foreground and background modulation, and how this may be accomplished (Huang et al., 2020; Papale et al., 2018; Scholte et al., 2008). Using fMRI, it has been shown that the frontal areas of the brain are involved in foreground and background modulation (Huang et al., 2020). However, the moment at which the frontal areas are involved in foreground and background modulation is not clear. Utilizing EEG, Scholte et al. (2008) found that there was a more negative amplitude in the posterior regions of the brain at earlier latency, but a more positive amplitude in the frontal electrodes after 208 ms, in response to figure textures as compared to uniform textures. The later positive response in the frontal electrodes is in line with the fMRI findings that the frontal areas are involved in texture segregation. However, this study focuses on the foreground processing of the figure textures. Future studies should focus on the involvement of high-level areas of the brain in foreground and background image processing in humans.

Previous studies have shown evidence of foreground enhancement and background suppression during texture segregation. However, when and how texture segregation involves in foreground and background modulation in the human brain is not well-understood. To reveal the temporal profile of foreground and background modulation during texture segregation, two major challenges must be addressed. First, the neural signals induced by the foreground should be separated from those of the background. Second, the neural activity should be recorded with sufficient temporal resolution in the whole brain during texture segregation. To overcome these challenges in the same recording in humans, we combined EEG recording and temporal response function (TRF) (Crosse et al., 2016; Lalor et al., 2006). The TRF method estimates the neural impulse response function evoked by certain stimuli to reflect the brain’s response to a unit change in the stimulus, such as a luminance change from ongoing visual stimuli (Lalor et al., 2006; Jia, Fang & Luo, 2019; Jia et al., 2017). It has been used to dissociate the neural correlates of individual auditory or visual streams from the mixture of concurrent streams (Crosse et al., 2016; Ding & Simon, 2012; Goncalves et al., 2014; Han & Luo, 2019; Jia, Fang & Luo, 2019; Jia et al., 2017; Liu et al., 2017).

The TRF method was used here to extract and dissociate the neural responses to the foreground and background during texture segregation. The neural responses specifically tracking the luminance sequence of the foreground and background were estimated from the EEG signal induced by ongoing figure textures. We used luminance-defined textures for two reasons. First, luminance-defined textures induce figure-ground modulation similar to orientation-defined textures in previous neurophysiological and EEG studies (Bach & Meigen, 1997; Zipser, Lamme & Schiller, 1996). Second, this satisfies the prerequisite of the TRF approach that a unit change in stimulus feature will induce a linearly varied EEG signal. The neural response to uniform texture was estimated from luminance changes of continuous homogeneous textures. It is worth noting that positive or negative deflections of the TRF do not necessarily map to enhancement or suppression. The foreground and background tsVEPs could then be extracted by subtracting the uniform TRF from the foreground and background TRFs. Previous EEG studies have found negative tsVEP components at posterior regions during texture segregation (Bach & Meigen, 1992, 1997; Heinrich, Andres & Bach, 2007). Based on combined evidence from neurophysiological (Klink et al., 2017; Lamme, 1995; Poort et al., 2016; Zipser, Lamme & Schiller, 1996) and neuroimaging (Kastner, De Weerd & Ungerleider, 2000; Papale et al., 2018; Poltoratski & Tong, 2020) studies showing foreground and background modulation in visual areas of the brain, we hypothesized that both negative components of foreground and background tsVEP may be found in the occipital regions. In addition, based on human imaging studies showing evidence of the involvement of high-level areas and positive component in anterior regions in texture segregation studies (Huang et al., 2020; Papale et al., 2018; Scholte et al., 2008), we hypothesized that the positive components of foreground and background tsVEP would also be observed in the anterior electrodes. In this study, we combined an EEG recording and TRF approach to investigate the time course involved in processing the foreground and background of an image during texture segregation in the whole brain.

Materials and Methods

Participants

Twenty-two adult participants (five males, age = 20.000 ± 1.826, age range = 17–25) were recruited for this study. All participants had normal or corrected-to-normal vision and reported no history of neurological disorders. All participants provided written informed consent prior to the start of the experiment, which was approved by the Committee on Human Research Protection of the School of Psychology of Shaanxi Normal University (ethical application ref: HR 2021-03-012).

Stimuli

The figure and uniform textures (34° × 25°) used in this experiment were generated by many randomly generated line elements (27 pixels long, 0.91°, and two pixels wide), as shown in Fig. 1. Specifically, the stimuli were made by randomly placing 13,000 gray line elements with a given orientation (45° or 135°) on a black background (1.86 cd/m2). The luminance of the textures ranged from 4.64 to 100.94 cd/m2 for each stimulus condition. The figure texture consisted of the foreground part and the background part. To make the figure appear stable, the foreground part (diameter: 8°) was made much smaller than the background part (the rest of the stimuli), since smaller regions are often (but not always) perceived as figures. The size of the stimulus would not affect the amplitude of the TRF response because in this experiment the stimulus size is constant throughout the trial (see more explanation in TRF calculation subsection). The figure textures were defined by the luminance difference between the foreground and the background, which was greater than 29.91 cd/m2, and the uniform textures were composed of lines of the same luminance. To enable TRF estimation, the luminance of the foreground and background was independently modulated in time at each frame (100 Hz) with the constraint that the minimal luminance difference between foreground and background was 29.91 cd/m2 for each frame. This value has been calculated by the group-wise threshold that was measured in a pilot psychophysical experiment (see Supplementary Information, Pilot Psychophysics Experiment 1). A novel temporal sequence was generated in each trial. The difference in the average luminance between the foreground (60.68 cd/m2) and background (60.64 cd/m2) were not significant (p = 0.575) at group level. Examples of figure texture trails have been made available for viewing (http://dx.doi.org/10.6084/m9.figshare.22434547).

Figure 1 Stimulus of experiment.

The luminance-defined figure and uniform texture are presented in the central visual field (A: figure texture; B: uniform texture).

Experimental procedure

All visual stimuli were generated using Psychtoolbox 3.0.14 (Kleiner et al., 2007) in the MATLAB environment. The stimuli were presented on a gamma-corrected CRT monitor (refresh rate: 100 Hz; resolution: 1,024 × 768; size: 17 inches) at a viewing distance of 57.3 cm. The head position of each participant was stabilized using a chin rest. A red fixation point was presented for 0.7–0.85 s, then a figure or uniform screen was presented for 5 s. The luminance of the texture was determined by luminance sequences of both the foreground and background. These were independently modulated throughout each trial at each frame between the darkest (4.64 cd/m2) to brightest gradients (100.94 cd/m2), as shown in Fig. 2A. Participants were asked to maintain their focus on a fixed point and to identify any changes in texture types throughout each trial. The participants were asked to identify whether the uniform texture suddenly presented itself in the figure texture trials or whether the figure texture appeared in the uniform texture trials. The different stimuli were presented for 0.5 s and occurred randomly between 0.25 and 4.25 s of the 5 s trial in 16.67% of the trials. Participants were not informed of the probability of the difference in advance. The participants were asked to press one of two buttons at the end of the trial to report whether they had detected the difference and the order of the buttons was balanced between participants. The inter-trial interval was 2 s. There were 126 trials for both the figure and uniform texture.

Figure 2 Psychophysical procedure and illustration of the temporal response function (TRF) approach.

(A) A sample trial sequence of experiment. (B) Three independent 5 s random temporal sequences were generated by modulating the luminance of the foreground, background, and uniform independently and randomly throughout the trial (e.g., left of B, top: foreground luminance sequence, middle: background luminance sequence, bottom: uniform luminance sequence). The EEG response was also recorded simultaneously (middle of B, top: figure EEG, bottom: uniform EEG). The TRF response for each stimulus condition (e.g. foreground, background and uniform) could be estimated from the EEG data based on the corresponding luminance temporal sequence (e.g., right of B, top: foreground TRF, middle: background TRF, bottom: uniform TRF). In order to compute the TRF, a regularized linear regression was applied between the stimulus luminance value and EEG amplitude.

EEG data acquisition and preprocessing

The electroencephalographic (EEG) data were recorded using a 64-channel EEG cap according to the extended international 10/20 system (Brain Products, Munich, Germany), with a FCz reference and a forehead ground electrode. Electrode impedances were kept below 5 kΩ. EEG data were recorded at a sampling rate of 1,000 Hz and stored on hard disk for later analysis.

EEG data were preprocessed using the EEGLAB toolbox (Delorme & Makeig, 2004). EEG data were offline re-referenced to the averaged mastoids (TP9 and TP10) and were band-pass filtered between 1 and 40 Hz using a Butterworth IIR filter with the order of two (Han & Luo, 2019). Independent components analysis was performed to identify and filter out eye-movements and artifact components. The remaining components were back-projected onto the EEG electrode space. The continuous EEG data were pre-separated into epochs from −1,000 to 7,000 ms around the onset of the stimuli, and then baseline-corrections (−1,000 ms~0) were performed. Epochs with incorrect responses and in which the EEG voltages exceeded the threshold of ±100 µV were excluded for further analysis. There were more than 82 trails for the remaining number of epochs for each condition. To estimate TRF, the EEG data were then downsampled to 100 Hz to match the sampling frequency of the stimulus sequence. To eliminate the possible contamination of the onset and offset response on the TRF estimation, epochs from 500 to 4,500 ms were extracted for further analysis (Jia et al., 2017) (Fig. S1). These preprocessed EEG epochs and their corresponding luminance sequences were then entered in the TRF calculations.

TRF calculation

The TRF response describes the mapping relationship between the stimulus input (i.e., the stimuli luminance sequence in the present experiment) to the brain and its output (i.e., the recorded EEG data) (Lalor et al., 2006). It characterizes the brain’s linear transformation of the continuously-changing stimulus to the ongoing neural activity. In brief, the TRF response is modeled as r(t,n)=∑τ⁡ω(τ,n)∗s(t−τ)+ε(t,n) where r(t, n) is the recorded EEG response at channels n at each time, s(t) is the stimulus property at each time, the symbol * denotes convolution, and ε is the residual response at each channel. ω(τ,n) is the to be estimated TRF of each channel for a range of time lags, τ, relative to the transient occurrence of stimulus feature, s(t). Given the known stimulus sequence and the recorded EEG signals, we were able to fit the TRF to the data. Specifically, the TRF response for each stimulus condition (e.g., foreground, background, and uniform) could be estimated from the EEG data based on the corresponding luminance temporal sequence (Fig. 2B). The estimated TRF response characterized the average neural response for each unit increment in the luminance sequence at each stimulus condition as a function of time lag relative to each transient occurrence of the stimulus feature.

The stimulus luminance sequences and EEG signals were concatenated across trials, respectively, and then normalized before TRF calculation. This was done for each condition, each channel, and in each subject separately. Then, using the multivariate temporal response function (mTRF) toolbox (Crosse et al., 2016), the TRF calculations were performed by a regularized linear regression (Fig. 2B), with the lambda parameter set to 1 for all subjects to control for overfitting. The TRF responses for uniform were extracted from the EEG recordings in uniform textures, while those for the foreground and background were calculated from the EEG data sets in figure textures, based on the corresponding luminance temporal sequences. The foreground- and background-tsVEP were then calculated by subtracting the uniform TRF from the foreground TRF and background TRF, respectively. These were performed for each stimulus condition, on each channel, and in each subject, respectively, then averaged across subjects.

The TRF approach has been used and validated in previous auditory and visual studies. Previous studies examined the brain responses which specifically tracked the stimulus modulated by the experimenter, such as a sound envelope (Ding & Simon, 2012; Lalor & Foxe, 2010), luminance sequence (VanRullen & Macdonald, 2012), or contrast sequence (Han & Luo, 2019). While the TRF approach is, in a sense, a generalization of the VEP, there are important differences between the TRF and the typical VEP approach (Crosse et al., 2016; Lalor et al., 2006). Notably, the TRF reflects the brain’s response to specific stimulus parameters defined by the experiment, not the entire stimuli. For example, the TRF response would not reflect the stimulus size because it is constant throughout the whole trial in this experiment (see the TRF response to small and large stimuli in Fig. S4). Moreover, the TRF has an advantage over VEP in that it can assess the brain’s response to the changing luminance of multiple simultaneous stimuli by applying independent stimuli sequences in single trial (Lalor et al., 2006; Liu et al., 2017). A previous EEG study showed that the TRF responses obtained in two simultaneous stimuli settings were comparable to those in a single stimulus presentation setup (Lalor et al., 2006). The results of that study indicate that TRF is capable of tagging multiple stimuli based on independent luminance sequences. Other studies have also been able to robustly track the luminance change of multiple simultaneous stimuli whose luminance were modulated independently for each stimulus (Huang et al., 2018; Jia, Fang & Luo, 2019; Jia et al., 2017). The TRF response to the foreground, background, and uniform are shown in Fig. 3. It is notable that the TRF response was flat and noisy (Fig. 4) when the EEG signals and stimulus sequences were shuffled across the trials (Jia, Fang & Luo, 2019; Jia et al., 2017). This further supports the fact that the calculated TRF waveform represented a stimulus-specific tracking response.

Figure 3 Original TRF waveforms.

The average (N = 22) TRF waveforms for the foreground (salmon pink lines), background (dark blue lines), and uniform (gray lines) were plotted in different brain regions (occipital, parieto-occipital, parietal, central, and frontal regions) as a function of temporal lag (0 to 800 ms).

Figure 4 TRFs obtained after the stimulus sequences and EEG were shuffled (B) or not shuffled (A).

Statistical analysis

The first step was to determine whether foreground and background modulation were present in each of the brain regions. The presence of negative component in the tsVEP was first assessed in the occipital region, as we expected to find the largest texture segregation responses here, according to previous tsVEP studies (Bach & Meigen, 1992; Caputo & Casco, 1999; Casco et al., 2005; Heinrich, Andres & Bach, 2007). To do this, the tsVEPs were calculated by algebraic subtraction of the uniform TRF from the foreground and background TRFs. The tsVEPs were used instead of the original VEP because the difference wave only reflected the processing specific to texture segregation rather than other confounding processing, such as the local processing of stimuli elements. Combined with latency and scalp distribution, the polarity of tsVEP component from previous studies, we determine the components of the current study. After visual inspection, the mean amplitude of early (100–140 ms) and late (250–320 ms) negative components was calculated, similar to the time window in previous tsVEP studies (Bach & Meigen, 1997; Heinrich, Andres & Bach, 2007). The mean amplitude of the positive component (150–200 ms) in the occipital region was calculated based on visual inspection. We assessed the presence of significant deflections in the time windows with one sample t-test (compared to 0) for the foreground and background conditions in the occipital region (O1, Oz, O2). Then, to check whether the deflection was also significant in other regions, one sample t-tests were also performed for parieto-occipital (POz, PO3, PO4), parietal (P1, Pz, P2), central (C1, Cz, C2), and frontal (F1, Fz, F2) regions for these components. Next, we examined whether the tsVEP was different between brain regions and stimulus conditions. Repeated-measures ANOVAs with stimuli conditions (foreground and background) and topographical factors (occipital, parieto-occipital, parietal, central, frontal) were performed separately for each component.

All statistical analyses were conducted using JASP 0.16.3.0 software (https://jasp-stats.org/) (Love et al., 2019). For each ANOVA, the sphericity assumption was assessed using Mauchly’s test. The Greenhouse-Geisser adjustment for non-sphericity was applied when appropriate. Post hoc tests were conducted when the interaction was significant. For all t tests, Cohen’s d effect size was calculated. The partial eta-squared ( ηp2) was reported to demonstrate the effect size in the ANOVA tests. P-values were corrected by Bonferroni adjustments to avoid multiple comparisons errors.

Results

Behavioral results

We used a signal detection method to analyze behavioral experiment data, and the hit rate and false alarm rate of the two conditions were assessed separately. Our results determined that the hit rate of the figure (M = 0.965, SE = 0.016) and the uniform (M = 0.972, SE = 0.015) were all higher than 0.92 (all t(21) > 2.885, p < 0.020, Cohen’s d > 0.614), and the false alarm rate of the figure (M = 0.048, SE = 0.012) and uniform (M = 0.010, SE = 0.003) were all lower than 0.08 (all t(21) < 2.755, p < 0.030, Cohen’s d < −0.587). The results showed that the participants perceived the figure and uniform textures, and performed the task well.

The early segregation component at 100–140 ms

The presence of foreground and background tsVEP

For the foreground tsVEP, as shown in Fig. 5, the early negative component was significantly in the occipital region (M = −0.006, SE = 0.002, t(21) = −3.561, p = 0.009, Cohen’s d = −0.759). For the background (Fig. 5), the mean amplitude in the early time window was significantly negatively deflected in the occipital and parieto-occipital regions (occipital: M = −0.016, SE = 0.003, t(21) = −5.167; p < 0.001, Cohen’s d = −1.102; parieto-occipital: M = −0.012, SE = 0.003, t(21) = −4.528, p < 0.001, Cohen’s d = −0.965), and the positive deflection was significantly in the frontal region (M = 0.004, SE = 0.002, t(21) = 2.915, p = 0.041, Cohen’s d = 0.622). The scalp topographies for the foreground tsVEP and background tsVEP are presented in Fig. 6.

Figure 5 Foreground tsVEP and background tsVEP waveforms in different regions.

Foreground tsVEP (foreground-uniform, red line) and background tsVEP (background-uniform, black line) waveforms in the different regions (occipital, parieto-occipital, parietal, central and frontal regions) as a function of latency (0–800 ms). Shaded regions represent analysis time windows (left to right: 100–140, 150–200, 250–320 ms). Red and black asterisks in the shaded regions indicate that the mean foreground and background tsVEP amplitudes deviated significantly from zero (p < 0.050) in the corresponding time windows, respectively.

Figure 6 Topographies of foreground tsVEP and background tsVEP for three components.

Grand average distribution maps for foreground tsVEP (top panel) and background tsVEP (bottom panel) at 100–140, 150–200, and 250–320 ms time range. White asterisks in occipital and frontal regions indicate foreground tsVEP or background tsVEP significantly different from zero (*p < 0.050, **p < 0.010, ***p < 0.001).

The modulation of texture segregation by stimuli and topography

To test whether the foreground- and background-related segmentation differed across brain regions, we performed a 2 × 5 (stimulus conditions: foreground, background × topographical factors: occipital, parieto-occipital, parietal, central, frontal) repeated measures ANOVA. Both the main effect of the topographical factors (F(4,84) = 20.754, p < 0.001, ηp2 = 0.497) and the interaction between stimulus conditions and topographical factors (F(4,84) = 13.574, p < 0.001, ηp2 = 0.393) were significant. Post hoc paired t tests showed that tsVEP was more negative in the background than the foreground in the occipital and parieto-occipital regions (occipital: difference = −0.011, SE = 0.003, F(1,21) = 12.008, p = 0.002, ηp2 = 0.364; parieto-occipital: difference = −0.008, SE = 0.003, F(1,21) = 9.289, p = 0.006, ηp2 = 0.307). However, these results were converse in the central and frontal regions (central: difference = 0.006, SE = 0.002, F(1,21) = 9.629, p = 0.005, ηp2 = 0.314; frontal: difference = 0.005, SE = 0.002, F(1,21) = 7.488, p = 0.012, ηp2 = 0.263), as shown in Fig. 7A. The other results were not significant (ps > 0.050). These results show that the occipital, parieto-occipital, parietal, and frontal regions are all involved in texture segregation.

Figure 7 Results for experiment: different patterns of foreground and background processing at 100–140 and 250–320 ms windows.

(A) Results for 100–140 ms window. (B) Results for 250–320 ms window. Error bars represent standard errors of the means. *p < 0.050, **p < 0.010, ***p < 0.001.

The positive component at 150–200 ms

The presence of foreground and background tsVEP

The early component in the 150–200 ms window was significantly positive in the occipital region for foreground tsVEP (M = 0.007, SE = 0.003, t(21) = 2.886, p = 0.044, Cohen’s d = 0.615), while significantly negative deflection was present in the central and frontal regions of the brain (central: M = −0.006, SE = 0.002, t(21) = −2.846, p = 0.048, Cohen’s d = −0.607; frontal: M = −0.005, SE = 0.002, t(21) = −2.933, p = 0.040, Cohen’s d = −0.625), as shown in Fig. 5. For the background tsVEP, significantly positive deflection was present in the occipital and parieto-occipital regions (occipital: M = 0.014, SE = 0.003, t(21) = 4.141, p = 0.002, Cohen’s d = 0.883; parieto-occipital: M = 0.012, SE = 0.003, t(21) = 3.953, p = 0.004, Cohen’s d = 0.843), while significantly negative deflection was detected in the central and frontal regions (central: M = −0.005, SE = 0.002, t(21) = −3.218, p = 0.021, Cohen’s d = −0.686; frontal: M = −0.006, SE = 0.001, t(21) = −4.026, p = 0.003, Cohen’s d = −0.858).

The modulation of texture segregation by stimuli and topography

To statistically test whether the different processing patterns between the foreground and background conditions were present in different regions, a two-way repeated measures ANOVA was performed with 2 (stimulus conditions: foreground, background) × 5 (topographical factors: occipital, parieto-occipital, parietal, central, frontal).The main effect of the topographical factors was significant (F(4,84) = 30.779, p < 0.001, ηp2 = 0.594), while neither the main effect of the stimulus conditions (F(1,21) = 1.350, p = 0.258, ηp2 = 0.060) nor the interaction between these factors (F(4,84) = 2.755, p = 0.094, ηp2 = 0.116) was significant.

The late segregation component at 250–320 ms

The presence of foreground and background tsVEP

As shown in Fig. 5, the late component in the 250–320 ms window was significantly positive in the central and frontal regions for foreground tsVEP (central: M = 0.005, SE = 0.001, t(21) = 3.362, p = 0.015, Cohen’s d = 0.717; frontal: M = 0.004, SE = 0.001, t(21) = 3.602, p = 0.008, Cohen’s d = 0.768). A significantly negative deflection was present in the occipital region (M = −0.004, SE = 0.001, t(21) = −2.867, p = 0.046, Cohen’s d = −0.611) for the background tsVEP (Fig. 5) while a significantly positive deflection was seen in the central and frontal regions (central: M = 0.005, SE = 0.001, t(21) = 3.137, p = 0.025, Cohen’s d = 0.669; frontal: M = 0.007, SE = 0.001, t(21) = 5.202; p < 0.001, Cohen’s d = 1.109).

The modulation of texture segregation by stimuli and topography

To statistically confirm the processing patterns between the foreground and background conditions across different regions, a two-way repeated measures ANOVA was conducted with 2 (stimulus conditions: foreground, background) × 5 (topographical factors: occipital, parieto-occipital, parietal, central, frontal). The main effect of the stimulus conditions was not significant (F(1,21) = 1.873, p = 0.186, ηp2 = 0.082), while both the main effect of the topographical factors (F(4,84) = 27.863, p < 0.001, ηp2 = 0.570) and the interaction between these factors (F(4,84) = 10.038, p < 0.001, ηp2 = 0.323) were significant. A simple effects analysis of the interaction showed that tsVEP was more negative in the background than the foreground in the occipital and parieto-occipital regions (occipital: difference = −0.005, SE = 0.002, F(1,21) = 8.340, p = 0.009, ηp2 = 0.284; parieto-occipital: difference = −0.005, SE = 0.002, F(1,21) = 8.293, p = 0.009, ηp2 = 0.283; all others p > 0.090) (Fig. 7B).

Discussion

We used EEG recordings and the TRF approach to investigate the spatio-temporal dynamics of figure-ground modulation in humans and to track the neural response of foreground and background processing from the overall figure-texture activity. We found negative components in the posterior regions and positive components in the anterior regions for both foreground and background part, but with different patterns of results. Specifically, in the posterior regions, we found an early negative component for foreground tsVEPs at 100–140 ms and two negative components for background tsVEPs at 100–140 ms and 250–320 ms. The different pattern of results in the posterior region of the brain suggests that the visual processing of the foreground and background varied at both the early and late phases. In the anterior regions, we found a later positive component of the foreground-tsVEP, and two positive components of the background-tsVEP at both early and late phases, which is consistent with previous studies on differential processing (Huang et al., 2020). In general, these results indicate that texture segregation involves both foreground and background modulation in both the early and late processing phases in humans.

The early segregation component at 100–140 ms window

Studies have found that the first negative component of the tsVEP response in the occipital region, which is more negative in the original VEP to figures than to uniform textures, was present at about 100 to 140 ms (Bach & Meigen, 1997; Caputo & Casco, 1999; Fahle et al., 2003; Heinrich, Andres & Bach, 2007; Lachapelle et al., 2004). This study incorporated and expanded on these findings and found that the first negative component of the tsVEP in the posterior region was present for both the foreground and background image processing at 100–140 ms. Based on the figure enhancement and background inhibition in neurophysiological (Klink et al., 2017; Poort et al., 2016) and brain imaging (Papale et al., 2018; Poltoratski & Tong, 2020) studies, we speculate that the foreground-related negative component may be related to figure enhancement, and the background-related negative component may be related to background inhibition. Previous studies have revealed that both foreground and background modulation may require early feedforward processing (Klink et al., 2017; Roelfsema et al., 2002), and we inferred that the early negative component in the posterior region may be related to feedforward visual processing. Moreover, we inferred that the early negative component found in previous tsVEP studies may have been jointly produced by the foreground and background parts of the figure. These results suggest that foreground and background modulation both are involved in texture segregation in the occipital regions during an early time window.

A number of studies have proposed two sub-processes for foreground enhancement: boundary detection (edge modulation) and region filling (center modulation) (Huang et al., 2020; Mumford et al., 1987; Poort et al., 2012; Roelfsema et al., 2002; Scholte et al., 2008). It is suggested that the boundary is detected through early feedforward processing within early visual areas, and the region is filled through later feedback processing (Knierim & van Essen, 1992; Lamme, Rodriguez-Rodriguez & Spekreijse, 1999; Poltoratski & Tong, 2020; Poort et al., 2012; Zipser, Lamme & Schiller, 1996). In one VEP study, it was found that the first negative component induced by a boundary was present around 100 ms and the region-filling related negative component was first detected around 170 ms (Scholte et al., 2008). The latency of the first negative tsVEP component in this study is similar to the latency of boundary detection in the previous study. We suspect that the early foreground-related negative component may mainly reflect the process of boundary detection. Nevertheless, we cannot deny a potential contribution from region filling to the first negative component based on latency coincidence. Previous studies have suggested that boundary detection is a pre-attentive process and is independent of awareness, while region filling requires attention and awareness (Caputo & Casco, 1999; Heinrich, Andres & Bach, 2007; Huang et al., 2020). Future studies should examine the influence of top-down information, like attention or awareness, on the first negative foreground-tsVEP component to differentiate between boundary detection and region filling.

Previous neurophysiological studies found that the initial latency of background inhibition at V1 differed from about 110 to 140 ms between studies (Chen et al., 2014; Poort et al., 2016). The present study found that the first negative background-tsVEP component in the posterior region occurred at about 100–140 ms, which is comparable to results from other studies (Chen et al., 2014). Given the evidence for background inhibition in these previous studies, we suggest that the negative component related to the background inhibition in the visual cortex. In addition, we observed an early positive component of background-tsVEP in the anterior regions, which have paid little attention in previous texture segregation EEG studies. The present of negative component in posterior region and positive component in anterior region may imply two sources for a background-related response, which is consistent with fMRI results (Huang et al., 2020) showing that both dorsolateral prefrontal cortex (DLPFC) and early visual cortex were involved in background suppression process. The prefrontal cortex is associated with the filtering of task-irrelevant distractors (Squire et al., 2013). Given that the task in the present study requires the participants to continuously monitor the texture stream to make judgments and thus need to filter out background signal. We speculate that the anterior positive component may be related to the current task. Future work is needed to test the specific role of the anterior brain regions (e.g., frontal) in background modulation using both task-relevant and task-irrelevant paradigms in the early process stage.

The late segregation component at 250–320 ms window

Some studies found the presence of a second negative component of tsVEP in the occipital region, that is, the amplitude of the overall figure-induced activity was more negative than uniform in the late window (Bach & Meigen, 1992; Casco et al., 2005; Heinrich, Andres & Bach, 2007). Our study extends these findings and suggests that the late negative segregation component in the occipital region was mainly induced by background texture. In most studies, the latest negative component of tsVEP in the occipital channels had shorter latencies than the latest negative component in the present study. Nevertheless, some studies also found a comparable latest negative component in the posterior region (Caputo & Casco, 1999; Schubo, Meinecke & Schroger, 2001). Neurophysiological studies found that background inhibition at V1 and V4 also continue into the late stage (Poort et al., 2016). These results are within the same window determined in our study. Based on background suppression evidence in neurophysiological and fMRI studies, we may infer that the latest negative component of background-tsVEP reflects the background suppression in the visual cortex.

The present study found that both the foreground and background related signals are expressed in the anterior regions during the late stage of texture segregation. Computational models (Mumford et al., 1987; Roelfsema et al., 2002) proposed that the region filling and background suppression require feedback processing from higher visual areas back to V1, which was supported by neurophysiological studies (Klink et al., 2017; Lamme, Zipser & Spekreijse, 1998; Poort et al., 2012; Self et al., 2013). The late emergence of the positive component in the anterior region was consistent with time consumption in the feedback process, which requires a longer time than feedforward processing. Its late occurrence provides enough time for feedback signals to be sent down to the early visual cortex. Neuroimaging studies in humans revealed that both the regional and background modulation at the V1 were driven from a feedback connection originating in the frontal area (Huang et al., 2020). We speculate that the late positive component in the anterior regions may relate to feedback processing from the frontal area, which is consistent with the proposed recurrent mechanism underlying texture segregation (Huang et al., 2020; Roelfsema et al., 2002). The precise origin of the positive component in the anterior region is not clear due to spatial resolution limitation. Further work using techniques with high spatial and temporal resolution (e.g., neurophysiological techniques or magnetoencephalogram) is also needed to examine how the high-level areas, like the frontal eye field (FEF) and DLPFC, contribute to texture segregation at different process stage.

Foreground-related processing is represented in the anterior regions of the brain in the late stage of processing, while background modulation is expressed in both the posterior and anterior regions. These results suggest that both the foreground and background modulation are involved in texture segregation in the latest phase.

Different patterns of foreground and background processing during figure-ground segregation

Foreground and background modulation require different computations and rely on different neural mechanisms (Poort et al., 2012; Roelfsema et al., 2002). Foreground enhancement involves boundary detection, which is achieved by feedforward processing within early visual cortex, and region filling, which requires feedback processing from higher visual areas to early visual cortex, whereas background suppression is completed by both feedforward and feedback connections between higher visual areas and early visual cortex. The present study found amplitude differences between foreground and background modulation in both the posterior and anterior regions, providing evidence that the processing pattern is different between foreground and background processing (Fig. 8). In order to create a clearer and more stable figure-ground perception, the foreground part of the texture stimuli was always presented centrally in the current study. This is in line with some previous studies (Casco et al., 2005; Poltoratski & Tong, 2020). Previous figure-ground segregation studies that used either central (Heinen, Jolij & Lamme, 2005; Poltoratski & Tong, 2020) or peripheral (Poltoratski et al., 2019; Scholte et al., 2008) figure stimuli found the same pattern of results pattern. Furthermore, one study used both central and peripheral foreground in their studies and found similar results, suggesting that the location of the figure stimulus does not affect the experimental results (Wokke, Scholte & Lamme, 2014). Therefore, it can be inferred that the different patterns of foreground and background modulation observed in the present study are not primarily due to the difference between central and peripheral stimulus presentation.

Figure 8 Schematic illustration of the figure-ground segregation.

First, image features, i.e., the local luminance and orientation of bars, are registered. Second, in the early stage of texture segregation, foreground enhancement and background suppression are expressed in the visual cortex, as evidenced by early negative components in the posterior region for both foreground- and background-tsVEPs. The positive component in the anterior region may be related to the process of filtering out the background signal in order to perform the detection task. Third, the background continues to be suppressed, which may be complete by feedback processing from the frontal areas to visual areas, as suggested by both the anterior positive component and the posterior negative component in the late time window. The positive component may be related to attention or other cognitive processes being directed to the figure-defined objects. The red dashed line represents foreground related processes, while the blue dashed line represents background related processes.

Consistent with previous studies demonstrating a texture segregation-related negative component in posterior regions (e.g. Caputo & Casco, 1999; Heinrich, Andres & Bach, 2007), our study further showed that both the foreground and background parts of the textures induced the negative component in the posterior regions. Moreover, the amplitude of the posterior component was more negative for background modulation than for foreground modulation at both early and late stages. The two negative components of the background-VEP in the posterior region imply that background suppression is completed by recurrent processing. Since the mean luminance of the foreground and background parts of the texture is not significantly different, the amplitude difference between them is not due to luminance difference. Similar to our study, evidence from neurophysiological (Poort et al., 2012) and brain imaging (Huang et al., 2020; Papale et al., 2018) studies suggest different processing patterns between foreground and background modulation, as enhancement and suppression, respectively, in early visual cortex. The polarity of the component cannot indicate neural enhancement or suppression, but the more negative response to background modulation than to foreground modulation suggests the different visual processing between these two processes, most likely, as foreground enhancement and background suppression in visual areas.

In the anterior brain regions, both the foreground and background parts of the texture induced positive components at different time windows. At the early stage, the amplitude of the background-tsVEP is larger than that of foreground-tsVEP, suggesting a different pattern of processing in the anterior region (e.g., frontal areas). As we discussed earlier, the early positive component might be related to the inhibition of the background as task-irrelevant distractor in frontal areas. At the late stage, there is no significant difference between the foreground- and the background-tsVEP. It seems that the foreground- and background-related processes are similar in the anterior region. However, different frontal areas are involved in foreground and background modulation (Huang et al., 2020). The frontal eye field (FEF), which is associated with the voluntary allocation of visual attention (Squire et al., 2013), is involved in foreground enhancement, whereas the DLPFC, which is related to the filtering of task-irrelevant distractors (MacDonald et al., 2000; Suzuki & Gottlieb, 2013), is involved in background suppression. We cannot exclude that the similar positive component between foreground and background may reflect different processes. One EEG study found the foreground-related positive component after 208–248 ms in the frontal area, which has been proposed to be related to cognitive processes such as directing attention to the texture-defined objects (Scholte et al., 2008). However, to our knowledge, no studies have investigated the background-related component. Nevertheless, further EEG work is needed to confirm the function of the anterior component by investigating the role of attention and/or task relevance on the anterior positive component of the foreground- and background-tsVEP at different time windows in figure-ground segregation.

The current study observed that both posterior (occipital, parieto-occipital) and anterior (frontal, central) brain regions processed foreground and background parts of the texture differently, consistent with a multiple brain regions mechanism for texture segregation. Through the interaction between visual areas and frontal areas, foreground and background are processed differently in different brain regions.

Conclusions

This study extracted and dissociated the foreground and background processing-related activity of the brain from a whole figure response using EEG recordings and the TRF approach. In the posterior region of the brain, the background-related segregation process induced negative tsVEP components in both the early and the late stages, while the foreground-related segregation process induced a negative component only in the early stage. In the anterior region of the brain, we found positive components in the early and late stages of processing background images. However, positive components were only found in the later stage of processing foreground images. The different processing patterns for foreground and background modulation jointly contribute to figure-ground texture segregation at both the early and late stages. Our findings support foreground-background modulation using a recurrent mechanism (Huang et al., 2020; Roelfsema et al., 2002; Scholte et al., 2008).

Supplemental Information

Supplemental Information 1 Supplementary figure and materials.

Click here for additional data file.

Additional Information and Declarations

Competing Interests

Author Contributions

Human Ethics

Data Availability

The authors declare that they have no competing interests.

Baoqiang Zhang conceived and designed the experiments, performed the experiments, analyzed the data, prepared figures and/or tables, authored or reviewed drafts of the article, and approved the final draft.

Saisai Hu conceived and designed the experiments, performed the experiments, authored or reviewed drafts of the article, and approved the final draft.

Tingkang Zhang conceived and designed the experiments, performed the experiments, authored or reviewed drafts of the article, and approved the final draft.

Min Hai conceived and designed the experiments, authored or reviewed drafts of the article, and approved the final draft.

Yongchun Wang conceived and designed the experiments, authored or reviewed drafts of the article, and approved the final draft.

Ya Li conceived and designed the experiments, analyzed the data, prepared figures and/or tables, authored or reviewed drafts of the article, and approved the final draft.

Yonghui Wang conceived and designed the experiments, prepared figures and/or tables, authored or reviewed drafts of the article, and approved the final draft.

The following information was supplied relating to ethical approvals (i.e., approving body and any reference numbers):

The Committee on Human Research Protection at the School of Psychology of Shaanxi Normal University granted ethical approval to carry out the study within its facilities (Ethical Application Ref: HR 2021-03-012).

The following information was supplied regarding data availability:

The original data and examples of stimulus presentation for this study are available at Figshare: Zhang, Baoqiang (2023). Different patterns of foreground and background processing contribute to texture segregation in humans: an electrophysiological study. figshare. Dataset. https://doi.org/10.6084/m9.figshare.22434547.v12.

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
