# Peer review of "Different patterns of foreground and background processing contribute to texture segregation in humans: an electrophysiological study"

_PeerJ, doi:10.7717/peerj.16139_

## Round 0.1 · original submission · Major Revisions

Both reviewers agree that this paper has the potential to fill a gap in knowledge. However, the hypothesis regarding the expected results and the interpretation of the results should be clarified. In addition, some of the reviewer comments regarding whether subjects experienced the expected percept must be addressed. Please address every comment from the reviewers in the revision.

Reviewer 1 ·

Basic reporting

• Clear, unambiguous, professional English language used throughout.
>> The use of English is not always as clear. Some sentences required a few re-readings to understand them because of grammar mistakes (missing words, singular/plural, etc.).

• Intro & background to show context. Literature well referenced & relevant.
>> Context is present, but a clear hypothesis or prediction is lacking which makes the interpretation of the complex data patterns difficult.

• Structure conforms to PeerJ standards, discipline norm, or improved for clarity.
>> Structure of the paper is fine

• Figures are relevant, high quality, well labelled & described.
>> The supplied labels are clear, but more figures could have helped interpretation.

• Raw data supplied (see PeerJ policy).
>> Raw(ish) data has been submitted for the behavior and TRF traces per subject. Raw EEG recordings before TRF analysis do not seem to be included.

Experimental design

• This is original primary research within the scope of the journal.

• Research question well defined, relevant & meaningful. It is stated how the research fills an identified knowledge gap.
>> The research is somewhat well-defined. It is clear that the authors aim to look at the temporal profile of figure-ground modulation in human EEG signals. However, the analysis is complex and results in a derived result signal in which statistics are calculated. A better prediction of what would constitute a meaningful result and how the data should be interpreted would be really helpful.

• Rigorous investigation performed to a high technical & ethical standard.
>> Some additional analysis/simulation is necessary to convince the reader that the method is indeed capable of independently extracting two signals (TRF) from the same EEG trace. Ethical documents associated with the study are provided.

• Methods are described with sufficient detail & information to replicate, but see above: additional checks, explanation, and prediction is necessary.

Validity of the findings

• Impact and novelty not assessed. Meaningful replication encouraged where rationale & benefit to literature is clearly stated.
>> The impact is difficult to assess since the meaning of the reported patterns of results is unclear. Statistical differences are found but how should tsVEP amplitudes, area differences, and temporal window differences be interpreted?

• All underlying data have been provided; they are robust, statistically sound, & controlled.
>> The data underlying the presented analysis has been provided, but not in their rawest form. TRFs and behavior is included. If I’m not mistaken, the raw or preprocessed EEG signals are not shared. The TRF method to extract meaningful signal seems crucial for the current study and only ‘raw’ data after this procedure is included.

• Conclusions are well stated, linked to original research question & limited to supporting results.
>> The conclusions are limited to a summary of the results and are consistent with the data, but they lack interpretation.

Additional comments

This study investigates the neural signatures of figure-ground segregation with EEG in humans. This phenomenon has extensively been studied in monkeys using electrophysiology and these authors set out to investigate whether the figure-enhancement with background-suppression seen in spiking data can also be observed with EEG. To this end, they have developed a technique to extract specific foreground and background responses from the same overall EEG signal using temporal response functions. With this technique they report finding different signatures for foregrounds and backgrounds that they relate to figure-ground segregation.

The authors claim that previous studies in humans did not measure background suppression because they lacked a neutral condition. The study by Papale et al. (2018) the authors cite in line 88 however does show foreground enhancement and background suppression in human visual cortex with fMRI. Whereas Papale et al. do use a different method than Poort et al. (NHP electrophysiology), and they lack the temporal information to dissociate early and late responses, they do show convincingly that backgrounds are suppressed in figure-ground segregation.

Why only luminance defined figure-ground in these texture stimuli? I guess this is required for the luminance fluctuation necessary for the TRF method, but it makes it a bit difficult to compare to previous studies. It would also be nice to provide an example movie of the stimulus as it is a bit difficult to imagine how strong the figure-ground percept is with this luminance manipulation.

‘The results showed that the participants maintained good fixation at the central fixation, and completed the task more seriously‘ (ln 238-239) >> This statement is supported by statistical analysis of hit-rate and false alarms. How does this relate to ‘good fixation’ or ‘serious task completion’?

The TRF for foreground and backgrounds is extracted from the same EEG signal. It is not entirely clear to me if these signals can be extracted independently. The authors do show that no patterned TRF remains in the stimulus-EEG relationship is shuffled, but I would also like to see some ground truth simulation data for the independent reconstruction of two mixed signals. This may be inherent in the method, but it’s crucial for the conclusions, and I think the reader needs to be taken by the hand a bit more to explain how reliable these results are. Comparing Fig 4 and 5 for instance, it looks like the TRF waves for figure and ground are strongly anti-correlated. Is this a consequence of the method or the brain’s encoding?

It is also not clear to me how we can make the leap from amplitude of the tsVEP to ‘enhancement’ and ‘suppression’. In Figure 4 I see deflections in the TRP that are largest for the uniform texture, yet this is ‘figure enhancement’? Altogether, a large set of results is reported (tsVEP for foreground and background and for different regions of the brain) but it remains unclear what the specific prediction would be, how differences and polarities need to be interpreted and what the conclusions are. A detailed hypothesis of tsVEP patterns under the assumption of figure enhancement and background suppression would greatly facilitate understanding of the results.

I have been looking at Fig 6 & 7 for quite some time, but I do not see a pattern that suggests figure enhancement and background suppression to me. I must admit I am not used to EEG data. I do agree that patterns across the brain look different for the early and late component, but this seems primarily driven by changes in the foreground tsVEP amplitude. I do not doubt that the indicated statistical differences are real, but what does it mean? We are looking at differences between foreground and background tsVEP across brain regions and across temporal windows (early vs late), but what does it mean if tsVEPs are positive or negative?

Reviewer 2 ·

Basic reporting

1\ The English language should be improved throughout the manuscript to ensure that an international audience can clearly understand the manuscript. Throughout the manuscript the current phrasing critically impairs smooth understanding. Some examples are lines 53 & 346 - but it is more of a general phenomenon.

Experimental design

2\ The approach of the authors to separate foreground and background processing is very interesting and addresses a gap in the existing literature. If indeed different processing characteristics for foreground and background were to be found, this would definitely enhance our understanding of the correlates of texture processing. The methods however, in particular the psychophysical setup, would benefit from a more detailed description. For example, it remains unclear how luminance sequences were modulated (uniformly?) and if thresholds were calibrated individually or groupwise. Further, the exact luminance differences between foreground and background are critical to the findings. It is probably important for differential foreground-background responses if luminance is brighter in the foreground or the background and exactly how large the difference is. Averaging over various different kinds of “feature differences” might blur a pattern of differential responses, I think.

Validity of the findings

3\ I have two major reservations about the current experimental setup and if it is capable of underpinning the conclusions authors draw. Assuming separating FG and BG processing with the TRF process works, the major confound of the presented results is that stimuli were not controlled for their respective sizes with the disc (“foreground”) being much smaller that the surrounding texture (“background”). The larger display size of BG over FG probably influences tsVEP amplitude in an unforeseeable way – intuitively, one would expect stronger negativities (tsVEPs) in BG simply due to the texture covering a larger part of the visual field. Irrespective of this conjecture being true, the study design must control the size of both textures and keep them constant in order to access amplitude differences between BG and FG.

4\ If this issue were resolved, another major difficulty of the psychophysical setup would remain. The study design must ensure that observers indeed perceive the disc as the foreground and the surround as the background. With, as I understand it, luminance sequences of both the disc and the surround being independently modulated (l. 148 f) the amount of difference between the two texture regions varies as well as which of the two regions seems more “striking”. This might very well influence observers’ perception of what the “figure” is and what the “ground”. This, however, is important to investigate figure-ground perception as opposed to mere texture segregation. Previous studies (e.g. Zipser, Lamme, & Schiller, JoNeuroscience, 1996) had similar stimulus material, but they investigated cell recordings in nonhuman primates, and were able to ensure that the receptive field of the cell covered either the central part or the background. To find a display that guarantees that each participant regards foreground and background as such while keeping display sizes of both at the same time constant is nontrivial. It might possibly help to keep one dimension (luminance of the background or luminance of the foreground) constant while only manipulating the respective other part? Or to define the figure by some other visual feature altogether (e.g. the orientation of the line elements) which is kept constant while luminance sequences are modulated, even though that might introduce unwanted summation effects between the two visual features.

---

## Round 0.2 · Major Revisions

The paper is improved but there are still a number of concerns that must be addressed. In particular, the reviewer is suggesting that the conclusions are overstated. The figures require clarification as described by the reviewer. Finally, please address the brain mechanism responsible for fig-ground segregation.

Reviewer 1 ·

Basic reporting

The reporting has become much clearer making the overall story much easier to follow. There are still a few language error but not as much. Structure has much improvement as well.

Experimental design

The design has been clarified, but a few questions remain to be discussed (see comments)

Validity of the findings

The findings seem ok, but the interpretation is sometime not entirely true to the data. Some rephrasing and additional discussion is warranted (see comments).

Additional comments

The paper has become much clearer and easier to follow. The TRF method is now explained much better and supported with references to other studies.

The main focus of the study seems to be two-fold:
1) Can we see modulation of the EEG signal specific for both/either figure and background?
2) What is the temporal profile of such modulation in anterior/posterior parts of the brain?

A uniform texture was used in NHP studies as well (Klink et al.; Poort et al.) but there, researchers have the advantage that the signals they record have a receptive field that either explicitly represents the figure or the background.

I think it can be made a little more clear that positive and negative deflections of the TRF signal do not necessarily map to enhancement or suppressions and that in fact they should be regarded more as a signature of ‘something is happening’ with no clear interpretation of its values.

One thing that may still require some discussion is the following: In the current experiment the luminance of figure and background were both manipulated at the same time. One might expect that if only the luminance of the figure OR the background was manipulated one would find the same signatures but only for those components. Because this is not done, a main concern is that the signatures that are now found and attribute to figure/background could also be attributed to the more simple fact that there are two sources of luminance fluctuations.

In addition, because the figure stimulus seems to have always been in the middle of the screen so instead of Fig/gnd the traces can also be attributed to central/peripheral stimulus information. So, are we really looking a figure-background segregation process? This warrants discussion.

I could not find any figure captions, so the interpretation of figures was a little unsure. For instance, I assume that the green TRFs of figure 3 represent uniform stimuli (as in fig 4)?

The stimulus movies on FigShare where helpful in understanding the experimental procedures. I think it might also be nice to indicate the luminance fluctuations in Fig 2A, perhaps be ‘stacking’ different luminance images in the stimulus period representation.

Please indicate in Fig 5 what the time windows are that were used for the statistical analysis and where the figure and background traces significantly deviated from the uniform TRF traces. This is arguably a more relevant statistical test than the comparison between figure and ground modulations than seem to be indicated. Of course, the latter comparison is also interesting but mostly after the presence of a modulation has been established.

What does it mean if the tsVEP amplitude is significantly different between Fig and gnd? Does it have a mechanistic interpretation?


It's unclear how significance is supposed to be inferred from figure 6. Might be in the missing captions?

Figure 7 and 8 can probably be 2 panels of the same figure because they essentially show the same type of data for different time windows.

I think it is an overstatement to say (390-392): “In general, these results indicate that texture segregation relies on both foreground and background modulation in both the early and late processing phases in humans.” It is not clear that it *relies* on it because no causality is shown. All that is shown is that there is a dissociable representation of foreground and background luminance (but see earlier comments).

The early component in anterior regions feels a little under-discussed. How does it fit theories? What I’m missing is a schematic of how the authors interpret their patterns of results in terms of a brain-wide fig-gnd segregation mechanism.

---

## Round 0.3 · Minor Revisions

The authors have addressed some of the comments from the reviewer, but not all of them. Please carefully address each of the latest comments from the reviewer in more detail. The reviewer also mentions grammatical errors which will also need to be addressed.

**Language Note:** The Academic Editor has identified that the English language must be improved. PeerJ can provide language editing services - please contact us at copyediting@peerj.com for pricing (be sure to provide your manuscript number and title). Alternatively, you should make your own arrangements to improve the language quality and provide details in your response letter. – PeerJ Staff

Reviewer 1 ·

Basic reporting

-

Experimental design

-

Validity of the findings

-

Additional comments

The authors have improved their manuscript. I especially appreciate the changes to the figures which have clarified some of my questions.

However, while some of my questions have been answered with this revision, others have not. For instance, where I commented that NHP neurophysiology studies have indeed looked at foreground and background modulation separately, the authors seem to acknowledge that in their reply, yet the manuscript still reads: “However, these studies were not able to determine whether fig-gnd modulation was caused by neuronal responses to the foreground, the background, or both.” I think this is simply incorrect. In studies where the neuronal receptive field only samples the foreground or background this can and has been determined (e.g., Klink et al.; Poort et al.). This is not a problem for the current study as it still nice that the authors can do it non-invasively in humans, but the claim that it is the first time this dissociation can be made is just wrong.

The authors have added an explanation that TRF deflections cannot be readily interpreted as enhancement or suppression to the methods and discussion. This is nice, but I strongly suggest they also mention it in the introduction when they first talk about the TRF method (around ln 130). It’s a bit lateto first mention this in ln 276.

I asked the authors to discuss the possibility of a central-peripheral difference as a potential confound for the signal differences attributed to fig-gnd segregation. They do not discuss this in the revised paper and in their reply say “We have reason to believe that the current experimental results really reflect the fig-gnd segregation process rather than the difference between central/peripheral stimulus information”. It’s great that the authors have these reasons, but please also make the argument for the reader.

Some of the added sentences have spelling or grammatical errors (mostly with the verbs). Not to the level that I cannot understand what is meant, but please check them carefully.

---

## Round 0.4 · accepted · Accept

The authors have responded adequately to the reviewer's latest comments. I do think the paper will need to be carefully proofread and examined for grammatical errors by the editorial staff prior to publication.